# Inflammatory Conditions Disrupt Constitutive Endothelial Cell Barrier Stabilization by Alleviating Autonomous Secretion of Sphingosine 1-Phosphate

**DOI:** 10.3390/cells9040928

**Published:** 2020-04-10

**Authors:** Jefri Jeya Paul, Cynthia Weigel, Tina Müller, Regine Heller, Sarah Spiegel, Markus H. Gräler

**Affiliations:** 1Department of Anesthesiology and Intensive Care Medicine, Jena University Hospital, 07740 Jena, Germany; jefri.jeyapaul@gmail.com (J.J.P.); Cynthia.Weigel@vcuhealth.org (C.W.); Tina.Mueller2@med.uni-jena.de (T.M.); 2Center for Molecular Biomedicine, Jena University Hospital, 07745 Jena, Germany; regine.heller@med.uni-jena.de; 3Center for Sepsis Control and Care, Jena University Hospital, 07740 Jena, Germany; 4Leibniz Institute on Aging—Fritz Lipmann Institute, 07745 Jena, Germany; 5Department of Biochemistry and Molecular Biology, Virginia Commonwealth University School of Medicine, Richmond, VA 23298, USA; sarah.spiegel@vcuhealth.org; 6Institute of Molecular Cell Biology, Jena University Hospital, 07745 Jena, Germany

**Keywords:** S1P receptor, inflammation, S1P transporter, spinster homolog 2, barrier dysfunction

## Abstract

The breakdown of the endothelial cell (EC) barrier contributes significantly to sepsis mortality. Sphingosine 1-phosphate (S1P) is one of the most effective EC barrier-stabilizing signaling molecules. Stabilization is mainly transduced via the S1P receptor type 1 (S1PR1). Here, we demonstrate that S1P was autonomously produced by ECs. S1P secretion was significantly higher in primary human umbilical vein endothelial cells (HUVEC) compared to the endothelial cell line EA.hy926. Constitutive barrier stability of HUVEC, but not EA.hy926, was significantly compromised by the S1PR1 antagonist W146 and by the anti-S1P antibody Sphingomab. HUVEC and EA.hy926 differed in the expression of the S1P-transporter Spns2, which allowed HUVEC, but not EA.hy926, to secrete S1P into the extracellular space. Spns2 deficient mice showed increased serum albumin leakage in bronchoalveolar lavage fluid (BALF). Lung ECs isolated from Spns2 deficient mice revealed increased leakage of fluorescein isothiocyanate (FITC) labeled dextran and decreased resistance in electric cell-substrate impedance sensing (ECIS) measurements. Spns2 was down-regulated in HUVEC after stimulation with pro-inflammatory cytokines and lipopolysaccharides (LPS), which contributed to destabilization of the EC barrier. Our work suggests a new mechanism for barrier integrity maintenance. Secretion of S1P by EC via Spns2 contributed to constitutive EC barrier maintenance, which was disrupted under inflammatory conditions via the down-regulation of the S1P-transporter Spns2.

## 1. Introduction

Endothelial cell (EC) barriers are important intercellular structures that regulate the movement of fluids and dissolved substances into tissues. Maintaining barrier function is particularly important at sites where fluids need to be efficiently separated from tissues such as the vasculature, lymph vessels, gut, brain, and lung [1]. Several different junctional complexes are involved in barrier formation, including tight and adherens junctions, gap junctions, and desmosomes [2]. Adherens junctions are formed by cadherins and nectins and provide a mechanical linker similar to zippers, while tight junctions are formed by claudins, occludin, and junctional adhesion molecules in the transmembrane regions and perform most of the sealing functions to prevent passage of fluids and molecules [3,4]. The bioactive sphingolipid signaling molecule sphingosine 1-phosphate (S1P) and its G protein-coupled receptor S1PR1 are critical mediators of adherens junction assembly [5]. The deletion of S1PR1 in ECs or the deletion of the two known S1P-producing sphingosine kinases (SphK1 and SphK2) in hematopoietic cells and ECs of adult mice result in severe disruption of the EC barrier [6,7,8]. Despite this apparent phenotype, the exact mechanism of barrier maintenance by S1P is still unknown. One of the most puzzling questions is of how high S1P concentrations in circulation that are sufficient to induce activation-induced internalization and desensitization of S1PR1 are able to constitutively maintain the vascular EC barrier. Two models were proposed as potential explanations [8]: (1) the static model postulating that there is constantly sufficient S1PR1 expression on the luminal cell surface of ECs even at high S1P concentrations, due to efficient receptor recycling and (2) the dynamic model suggesting that S1PR1 is only expressed on the tissue-facing side of vascular ECs which are activated by S1P leaking through the EC barrier and subsequently induce adherens junction assembly and EC barrier stabilization. In either case, reduced S1P leakage and decreased barrier stability occur until the amount of S1P leaking through the EC barrier increases again and starts a new cycle of EC barrier formation. The validity of either of these models has not yet been verified.

The collapse of the EC barrier is a life-threatening condition and a major severity factor in sepsis [9]. The S1P concentration in circulation decreases significantly during systemic inflammation [10,11,12,13]. Whether or not this observed decrease of S1P has something to do with the vascular EC barrier collapse is not known. Previous data indicate that even low amounts of S1P in plasma are sufficient to maintain S1P and S1PR1 mediated lymphocyte circulation [14].

Here, we show that, in vitro, ECs can autonomously produce and secrete S1P, rendering their ability to maintain EC barrier formation largely independent from exogenously added S1P. The S1P transporter Spinster homolog 2 (Spns2) plays a crucial role in the proper release of S1P into the extracellular space. Our work has uncovered an important function of Spns2 in ECs to regulate barrier stability. Spns2 deficient mice demonstrated significantly reduced EC barrier formation presumably due to the lack of S1P exportation from ECs. Furthermore Spns2, but not S1PR1, was down-regulated in ECs stimulated with lipopolysaccharides (LPS) and pro-inflammatory cytokines. Inflammation-induced EC barrier breakdown due to down-regulation of Spns2 resulted in decreased S1P release. Thus, decreased exportation of S1P from ECs due to reduced expression of Spns2 may contribute to EC barrier dysfunction during inflammation. This mechanism may be particularly important in sepsis, where inflammation-induced collapse of the EC barrier significantly contributes to increased morbidity and mortality. The observed stable expression of S1PR1 and the most likely local autocrine and paracrine activity of Spns2-released S1P points to local approaches for S1P supplementation in tissues rather than systemic alteration of S1P in circulating plasma as a potential medical intervention.

## 2. Materials and Methods

### 2.1. Cell Culture

Human umbilical vein EA.hy926 cells (ATCC CRL-2922) were grown in M199 medium containing 10% fetal bovine serum (FBS; Biochrom, Berlin, Germany), 1% penicillin/streptomycin (100 U/mL, Biochrom), 0.2% glutamine (200 mM, Lonza, Basel, Switzerland), 0.2% heparin (12.5 mg/mL, Carl Roth, Karlsruhe, Germany), and 0.6% ascorbic acid (20 mM, Sigma-Aldrich, Steinheim, Germany). HUVEC were freshly isolated from human umbilical cords and grown in M199 containing 17.5% FBS, 1% penicillin/streptomycin (100 U/mL), 0.34% glutamine (200 mM), 0.2% heparin (12.5 mg/mL), 0.5% ascorbic acid (20 mM) and endothelial mitogen (5 mg/mL, Alfa Aesar, Karlsruhe, Germany). FBS was heat inactivated at 56 °C. Rat hepatoma HTC4 cells expressing human S1PR1 together with human Gαi subunit of trimeric G proteins [15] were grown in minimal essential medium (MEM) with Earle’s salts (MEM Earle’s medium) containing 10% FBS, 2% 100× non-essential amino acids (Biochrom), 1% 100 mM sodium pyruvate (Biochrom) and 1% penicillin/streptomycin (100 U/mL). Cells were incubated at 37 °C and 5% CO_2_ in a humidified incubator (Panasonic, Hamburg, Germany).

### 2.2. Isolation of Primary Lung Endothelial Cells

Mice deficient for the S1P-transporter Spinster homolog 2 (Spns2) and their wild-type (wt) littermates were obtained from the NIH Knockout Mouse Project [16]. All described animal procedures were done with dead mice in accordance with the Association for Assessment and Accreditation of Laboratory Animal Care (AAALAC), Animal Welfare Assurance Number AD10000996, effective date March 26, 2017, renewed February 21, 2020 at the Virginia Commonwealth University School of Medicine, Richmond, VA, USA. Mice were killed, and after perfusion through the left ventricle with PBS supplemented with 0.1% heparin (10 U/mL), lungs were dissected from the thoracic cavity and cut into single lobes. The bronchial area was removed and the remaining tissue was digested in 15 mL conical tubes containing 5 mL of PBS with 0.5 mg/mL Liberase TL (Sigma-Aldrich) and 0.02 mg/mL DNAse 1 (Thermo Scientific, Braunschweig, Germany) for 45 min at 37 °C with shaking. The tissue was subsequently minced, further incubated for 30 min and passed through a 70 µm cell strainer (BD Biosciences, Heidelberg, Germany). Single cell suspensions were centrifuged at 300 rcf for 10 min at 4 °C. Supernatants were aspirated and cell pellets re-suspended in 90 µL of PBS containing 2 mM EDTA, 0.5% BSA, and 10 µL of CD31 MicroBeads per 107 cells (Miltenyi Biotec, Bergisch-Gladbach, Germany). The suspensions were incubated for 15 min at 4 °C and re-suspended in 500 µL PBS with 2 mM EDTA and 0.5% BSA. The labeled cells were separated with LS columns and the QuadroMACS separation system according to the manufacturer’s instructions (Miltenyi Biotec). The isolated ECs were cultured in endothelial growth medium (5 × 105 cells/mL, Cell Biologics M1168) on poly-l-lysine-coated six-well plates.

### 2.3. cDNA Synthesis and Quantitative Polymerase Chain Reaction (qPCR)

Total RNA was extracted using the Quick-RNA Miniprep Plus kit (Zymo Research, Freiburg, Germany). cDNA was synthesized with the RevertAid first strand cDNA synthesis kit (Thermo Scientific) according to the manufacturer’s instructions and diluted in nuclease-free water to a final concentration of 5 ng/µL. qPCR was performed with 8 µL cDNA, 4.4 µL of 25 mM MgCl_2_ (VWR), 2 µL of 10× reaction buffer (VWR) with 15 mM MgCl_2_, 0.2 µL of 10 mM dNTPs (Thermo Scientific), 0.8 µL of 5 µM 6-carboxy-X-rhodamine (ROX, Eurofins, Ebersberg, Germany), 0.048 µL of 5 U/µL of Taq-Polymerase (VWR), 3.752 µL of nuclease free water (Thermo Scientific), and 0.8 µL of TaqMan primer/probe-mix (2.5 µM each, Eurofins) for each reaction. The reaction was performed with the Mastercycler Realplex (Eppendorf, Hamburg, Germany) using the following program: initial activation at 94 °C for 10 min followed by 40 amplification cycles of denaturation at 94 °C for 10 sec and annealing and extension at 60 °C for 1 min. The internal reference dye ROX was used to normalize the fluorescent reporter signal. The expressions of genes of interest were normalized to hypoxanthine-guanine phosphoribosyltransferase (*HPRT*). Relative gene expression was calculated by the ∆Ct and 2^−∆∆Ct^ methods [17]. Primers used for the reaction are listed in Table 1.

### 2.4. Agarose Gel Electrophoresis

Gel electrophoresis was performed with 1.2% agarose gels in TBE buffer according to standard protocols for 30 min at 150 V. Gels were stained with ethidium bromide and visualized with a UV trans-illuminator.

### 2.5. Flow Cytometry

Cells were suspended in 200 μL of 5% FBS in PBS and transferred to a 96-well round bottom plate. The plate was centrifuged at 265 rcf for 5 min. After centrifugation, 300 μL of the primary antibody specific for the human S1PR1 receptor produced in mouse (20 µg/mL, custom-made by Abmart, Berkeley Heights, NJ, USA) was added and incubated for 1 h at 4 °C. After incubation, cells were centrifuged and washed with 5% FBS in PBS, followed by 30 min incubation with 50 μL of anti-mouse Biotin-SP (1:200 dilution, Jackson Immuno Research, West Grove, PA, USA, 115–065–075). Subsequently, cells were washed and incubated with 50 μL of 40 ng/mL streptavidin-PE (Biolegend, San Diego, CA, USA, 405203) for 30 min. Cells were washed again and resuspended in 200 μL of 5% FBS and 1 µg/mL propidium iodide (BD Biosciences) in PBS. Resuspended cells were analyzed with the Accuri C6 Plus (BD Biosciences). After doublet exclusion and life-dead discrimination by propidium iodid, the mean floerescence intensity was analyzed.

### 2.6. Western Blot and Calcium Measurement

Cells were washed and lysed with buffer containing radioimmunoprecipitation assay buffer (RIPA, Merck, Darmstadt, Germany), 0.5 M EDTA, and protease and phosphatase inhibitor cocktail (Thermo Scientific). Samples were adjusted to 10 µg protein with the BCA protein assay kit (Thermo Scientific) and blotted to polyvinylidenfluoride (PVDF) membranes (GE Healthcare Life Sciences, Freiburg, Germany) according to standard protocols. Detection was performed with antibodies against GAPDH (1:2000 dilution, Santa Cruz Biotechnology, Dallas, TX, USA SC-166574) and VE-cadherin (1:1000 dilution, BD Biosciences 610252), anti-mouse HRP secondary antibody (1:1000 dilution, Carl Roth 47591) and SuperSignal West Pico Chemiluminescent Substrate (Thermo Scientific) with the C-Digit Blot Scanner (LI-COR Biosciences, Bad Homburg, Germany).

Calcium measurements were performed as described [18].

### 2.7. Electric Cell-Substrate Impedance Sensing (ECIS)

ECIS arrays (Ibidi, Gräfelfing, Germany 8W10E PET) were coated with 200 µL of 0.2% gelatin (Sigma-Aldrich) for 20 min at room temperature. Gelatin was replaced with 400 µL of medium containing 1 × 105 cells. Measurements were carried out at 6 kHz with 10 s interval. Once a monolayer was attained, 200 µL of the medium was replaced by 200 µL of starvation medium (2% FBS containing growth medium). The monolayer was stimulated 6 h later with various stimulants. Results were analyzed using ECIS software v1.2.214.0 (Ibidi) by normalizing the data points after treatment to the data point before treatment. The basal barrier function was set to 1.

### 2.8. Fluorescence Microscopy

Cells were fixed with 100% ice-cold methanol for 15 min at −20 °C. Coverslips were washed five times by dipping them in ice-cold HBSS containing Ca^2+^/Mg^2+^. Slides were blocked in 50 µL blocking buffer (5% goat serum in HBSS -including Ca^2+^/Mg^2+^ and 0.1% saponin) for 1 h at room temperature followed by overnight incubation with 40 µL of mouse primary VE-cadherin antibody (BD Biosciences 610252) diluted 1:100 in 1% BSA in HBSS+ Ca^2+^/Mg^2+^ +0.1% saponin at 4 °C in a dark wet chamber. Slides were washed three times in washing buffer (0.1% saponin in HBSS containing Ca^2+^/Mg^2+^) and incubated in 40 µL of 10 µg/mL goat anti-mouse Cy3 secondary antibody (Life Technologies, Darmstadt, Germany A10521) diluted in 1% BSA in HBSS+ Ca^2+^/Mg^2+^ +0.1% saponin at room temperature in a dark wet chamber. Slides were washed and stained with 40 µL of 300 nM DAPI (VWR) for 20 min at room temperature. Then, the slides were washed three times in ice-cold PBS and mounted using CC Mount (Sigma-Aldrich).

### 2.9. Measurement of S1P and Sphingosine

Lipid extraction and quantification of S1P and sphingosine by liquid chromatography coupled to triple-quadrupole mass spectrometry (LC-MS/MS) was done as described [19].

### 2.10. FITC-Dextran Leakage Assay

Sixty thousand primary lung ECs in 100 µL medium were seeded in the upper chamber of Transwell inserts with 0.4 µm pores (Sarstedt, Nürnbrecht, Germany) in 24-well plates and grown until confluency. The lower chamber was filled with 600 µL medium. Afterwards, the media of the upper chamber was replaced with 200 µL of media containing 2 mg/mL 70 kDa FITC-dextran (Sigma-Aldrich). The amount of FITC-dextran was measured in the medium of the lower chamber after 24 h incubation at 37 °C and 5% CO_2_ with the Infinite 200 plate reader (Tecan, Stadt Crailsheim, Germany) at 485 nm excitation and 530 nm emission wave lengths. The exact amount of FITC-dextran was determined using a standard curve of FITC-dextran diluted in cell culture media.

### 2.11. Albumin Measurement

The detection of mouse albumin in bronchoalveolar lavage fluid (BALF) was carried out with the Mouse Albumin ELISA Quantitation Set (Bethyl Laboratories, Montgomery, TX, USA) according to the manufacturer´s instructions. BALF was diluted 1:20.000 in dilution buffer (50 mM Tris base, 0.14 M NaCl, 1% BSA, 0.05% Tween 20), and 100 µL were added to anti-mouse albumin antibody pre-coated 96-well plates.

### 2.12. Reagents

Stimuli used in this study were S1P (Sigma-Aldrich), FTY720 (Cayman Chemicals, Ann Arbor, MI, USA), FTY720-phosphate (Cayman Chemicals), W146 (Tocris, Wiesbaden-Nordenstadt, Germany), LPS (Sigma-Aldrich), TNFα (ImmunoTools, Friesoythe, Germany), and IL1β (Thermo Scientific). The anti-S1P antibody Sphingomab (LT1002) and the corresponding isotype control antibody LT1013 were kindly provided by Roger Sabbadini (LPath Inc. and San Diego State University, San Diego, CA, USA).

### 2.13. Statistics

Statistical analysis was performed using GraphPad Prism^®^ Software Version 5.00 (San Diego, CA, USA). Data are presented as mean ± SEM. Unpaired two-tailed t-tests were used to compare two groups. The significance threshold was set to * *p* < 0.05, ** *p* < 0.01, and *** *p* < 0.001.

## 3. Results

### 3.1. EC Barrier Stabilizing Function of S1P and S1PR1

To investigate the role of S1P in EC barrier function, the human endothelial cell line EA.hy926 and primary HUVEC were used. EA.hy926 represents a somatic cell hybrid of HUVEC and the lung epithelial carcinoma cell line A549. Quantitative PCR demonstrated that both, HUVEC and EA.hy926 expressed mainly *S1PR1* followed by *S1PR3*, although HUVEC expressed both receptors stronger than EA.hy926 (Figure 1A). Higher expression of S1PR1 in HUVEC was confirmed by flow cytometry (FACS) using a highly specific antibody against human S1PR1 (Figure 1B). Specific staining was demonstrated by the incubation of cells with 1 µM FTY720 overnight, which led to S1PR1 internalization and consequently low cell surface staining as expected (Figure 1B). In line with these expression data, HUVEC responses were greater than those of EA.hy926 after stimulation with 100 nM S1P in intracellular calcium flux measurements (Figure 1C). However, to our surprise, EA.hy926 responses were greater in ECIS measurements after stimulation with 1 µM S1P compared to HUVEC, indicating a higher barrier stabilization in EA.hy926 (Figure 1D). Basal resistance of the EC monolayer, however, was lower in EA.hy926 compared to HUVEC (Figure 1E).

### 3.2. Endogenous Differences in S1P Signaling between HUVEC and EA.hy926

To explore the reason for the different behavior of HUVEC and EA.hy926 in ECIS measurements, both cells were treated with 3 µM of the S1PR1 antagonist W146. While EA.hy926 resistance was not affected by W146 treatment, HUVEC monolayers showed significantly reduced resistance by 60% in ECIS measurements, suggesting involvement of S1PR1 in constitutive basal EC barrier maintenance in HUVEC, but not in EA.hy926 (Figure 2A). A similar observation was recorded in ECIS measurements after treatment with the anti-S1P antibody Sphingomab. Sphingomab (120 µg/mL) reduced the basal resistance of the HUVEC monolayer by 30%, while EA.hy926 did not respond at all (Figure 2B). Determination of S1P in the supernatant of both cell types revealed three fold greater S1P level in HUVEC medium than EA.hy926 medium (Figure 2C). Conditioned HUVEC medium consequently provided a four-fold enhanced calcium signal in S1PR1, overexpressing rat hepatoma HTC_4_ cells compared to EA.hy926 conditioned medium (Figure 2D). Conditioned medium from HUVEC induced a significant 20% increase of the measured resistance in ECIS experiments when added to EA.hy926, while conditioned medium from EA.hy926 in contrast reduced the corresponding resistance by 20% of a HUVEC monolayer (Figure 2E). HUVEC re-established their barrier integrity within hours, while the observed increased resistance in EA.hy926 after incubation with conditioned medium from HUVEC subsequently decreased further and fell below the value of HUVEC (Figure 2E).

### 3.3. Reversibility and VE-Cadherin Disturbance of EC Barrier Destabilization by S1PR1 Antagonism and S1P Blocking

Since HUVEC responded immediately to treatment with the S1PR1 antagonist W146 and the anti-S1P antibody Sphingomab with EC barrier disruption, we next asked if this destabilizing effect was reversible. To this end, HUVECs were applied to ECIS measurements and treated with either 3 µM W146 or 120 µg/mL Sphingomab. After decreased resistance leveled off in ECIS measurements, the medium was replaced without the addition of W146 and Sphingomab. Subsequently, the recorded resistance increased immediately and eventually reached normal levels of untreated control cells (Figure 3A). This experiment demonstrated that S1PR1 had to be constantly and persistently activated to induce the EC barrier stabilizing effect, and S1P had to be present all the time as a stimulus. Barrier stabilization was not induced by a single long-lasting activation of S1PR1, but required continuous stimulation. This was also supported by staining HUVEC monolayers for VE-cadherin expression. HUVEC showed pronounced VE-cadherin staining in the intercellular regions of monolayers, which was severely disrupted after treatment with 3 µM W146 (Figure 3B and Appendix A).

### 3.4. Role of Spns2 in EC Barrier Maintenance

Autonomously produced S1P by HUVEC, but not by EA.hy926 obviously contributed to constitutive basal EC barrier maintenance. qPCR data revealed that HUVEC expressed significant amounts of the S1P transporter Spns2 mRNA, while EA.hy926 were negative (Figure 4A). Monolayers of primary lung EC isolated from Spns2 deficient mice showed significantly decreased resistance values in ECIS measurements compared to those isolated from wt mice (Figure 4B). Spns2 deficient mouse lung EC also demonstrated increased leakage of fluorescein isothiocyanate (FITC) labeled dextran compared to wt mouse lung EC in Transwell cell monolayer permeability assays (Figure 4C). These results supported a significant contribution of Spns2-driven export of autonomously produced S1P for EC barrier formation ex vivo. To examine whether Spns2 also contributed to EC barrier stabilization in vivo, serum albumin leakage from circulation into the lung was measured in bronchoalveolar lavage fluid (BALF) of wt and Spns2 deficient mice. In line with ex vivo data, BALF retrieved from Spns2 deficient mice contained significantly more serum albumin compared to wt mice (Figure 4D).

### 3.5. S1P-Mediated EC Barrier Maintenance Under Inflammatory Conditions

To test the potential influence of inflammation on EC barrier formation, HUVEC and EA.hy926 monolayers were treated with a mix of the pro-inflammatory cytokines tumor necrosis factor-alpha (TNF-α) and interleukin-1beta (IL-1β) together with lipopolysaccharide (LPS). While HUVEC responded with a severe decrease of resistance in ECIS measurements upon treatment, EA.hy926 only showed a weak response (Figure 5A). S1P measurements demonstrated increased S1P levels in the supernatant of HUVEC, but not EA.hy926, which was markedly reduced after treatment with cytokines and LPS (Figure 5B). Conditioned HUVEC medium consequently reduced the calcium signal by 60% in S1PR1 overexpressing rat hepatoma HTC_4_ cells after treatment with cytokines and LPS compared to non-treated HUVEC medium (Figure 5C). qPCR analyses revealed decreased expression of Spns2 in HUVEC after cytokine and LPS treatment, while expression of the S1P-degrading enzyme S1P-lyase (SGPL1), the S1P-dephosphorylating enzyme lipid phosphate phosphatase 3 (LPP3), and SphK1 was increased (Figure 5D). SphK2 and S1PR1 expression did not change (data not shown). Compared to EA.hy926, HUVEC showed pronounced VE-cadherin staining in the intercellular regions of monolayers, which was severely disrupted after treatment with cytokines and LPS (Figure 6A). In contrast to EA.hy926, VE-cadherin protein expression was reduced after treatment with cytokines and LPS (Figure 6B). Importantly, S1P was able to substantially rescue barrier disruption of HUVEC monolayer in ECIS measurements by transiently increasing the resistance (Figure 6C).

## 4. Discussion

S1P is present in the circulation at high nM up to µM concentrations. Recent studies suggested that S1P in blood is particularly important for EC barrier maintenance [8]. Data presented in this study, however, demonstrate that autonomously produced S1P by EC is important for basal constitutive maintenance of barrier function. This is also consistent with previous studies that focused on the role of S1P in the circulation, but used inducible knockout strategies that would also delete the only S1P-producing enzymes SphK1 and SphK2 in EC [20]. Deletion of both SphK1 and SphK2 in EC prevented the autonomous production and release of S1P in EC. Furthermore, the important role of autonomously produced S1P in EC was previously demonstrated by the treatment of HUVEC monolayer with the S1PR1 antagonist W146 prior to addition of platelets in transendothelial electrical resistance (TEER) measurements, which also induced a significant reduction of resistance [20]. While the role of S1PR1 in EC barrier formation and maintenance was investigated many times, the source of S1P required for S1PR1 mediated EC barrier stabilization is still under debate. Our data support the autonomous contribution of EC to produce their own barrier-stabilizing S1P.

S1P is produced intracellularly and needs to be transported out of the cell to act as an extracellular ligand for S1PRs. The main S1P transporter expressed in EC is Spns2 [21,22]. Our data indicate that Spns2-deficient EC suffer from a compromised barrier function due to the defective export of S1P and low extracellular S1P concentrations, which resembled the decreased circulating levels of S1P in mice lacking Spns2. Isolated primary lung EC established very quickly a stable resistance base line as a measure for barrier integrity, lacking the common slow increase in resistance in the first couple of hours of this assay that we observed with EA.hy926 and HUVEC. One reason for this unusual behavior could be the lack of cell growth and division, and seeding of these cells in quantities sufficient to rapidly establish a confluent cell layer to compensate for their observed growth arrest. EC barrier destabilization in Spns2 deficient mice was not observed in a previous study using Evans Blue dye to investigate EC barrier leakage [16]. A possible reason could be different experimental approaches. In this study, we found that endogenous serum albumin was significantly increased in BALF of Spns2-deleted mice compared to wt mice, which reflects the steady-state of barrier stability formed over a long period of time. In contrast, Evans Blue leakage is a single event monitored over a very brief period of 90 min, which may not be sufficient to detect differences in basal EC barrier disturbances. Our ex vivo data was consistent with our in vivo data obtained in three different experimental setups with primary lung EC that all confirmed a disturbed EC barrier function after depletion of Spns2.

Deficiency of S1P in circulation contributes to detrimental effects during inflammation [8,20]. Although the EC barrier stabilizing function of S1P is well accepted, a contributing role of S1P production, transportation, and signaling in inflammation-induced EC barrier breakdown is a novel observation of this study. Particularly, the demonstration that cytokine and LPS-induced down-regulation of Spns2 in EC contributed significantly to lower basal extracellular S1P levels and consequent EC barrier disruption, has not been reported before. Thus far, many studies, including ours, observed an up-regulation of SphK1 during infection, and it therefore could be considered to be an inflammatory kinase [23,24]. The down-regulation of Spns2 in EC might be an effective response to support the infiltration of leukocytes at sites of local infection. In the event of a systemic infection, however, reduced expression of Spns2 in endothelial cells likely contributes to a global collapse of the vascular EC barrier leading to septic shock. Treatment of ECs with inflammatory stimuli did not compromise the expression of S1PR1 on EC, which opens the possibility of using S1PR1 agonists for EC barrier stabilization. Previous studies investigated the influence of S1PR1 agonists on EC barrier protection in various different disease models with mixed results. While some studies showed beneficial effects [7,25], others reported they were ineffective [26]. Thus far, even successful approaches with S1PR1 agonists were not very effective. These results are consistent with our data, which showed incomplete rescue of the EC barrier under inflammatory conditions, probably due to concomitant cytokine and LPS-induced down-regulation of VE-cadherin, which is an essential player in EC barrier formation as well [3].

The contribution of autonomously produced and secreted S1P by EC also implicates S1PR1 and EC barrier function. Since levels of S1P released by EC with 4–15 nM are much lower than the typical concentrations measured in plasma with 200–1000 nM, it is unlikely that activation occurs at the plasma-facing side of EC. EC barrier stabilizing stimulation supposedly occurs at the tissue-facing side of EC, and strategies to increase S1P in tissues such as inhibition of the S1P-metabolizing enzyme SGPL1 may be more promising for future medical interventions than gross application of S1PR1 agonists [27]. SGPL1 inhibitors have already been tested in rheumatoid arthritis [28]. Additional studies are required to evaluate the full potential of targeting S1P signaling and metabolism for EC barrier stabilization.

## 5. Conclusions

We confirmed that S1P is a major EC barrier-stabilizing factor, predominantly via S1PR1 stimulation. The constitutive production of S1P by EC and its release in the local environment by Spns2 rather than systemic S1P-levels in plasma were important for basal EC barrier stabilization. Inflammatory stimuli resulted in EC barrier disruption due to the down-regulation of the S1P-transporter Spns2, while the expression of S1PR1 was not altered. Based on our data, inflammation-induced EC barrier disruption may be prevented by local application of S1P or S1PR1 agonists, e.g., via permanent inhalation in the lung or by inhibition of cellular S1P-degradation in tissues with S1P-lyase inhibitors to compensate for the reduced release of S1P from EC.

## Figures and Tables

**Figure 1 cells-09-00928-f001:**
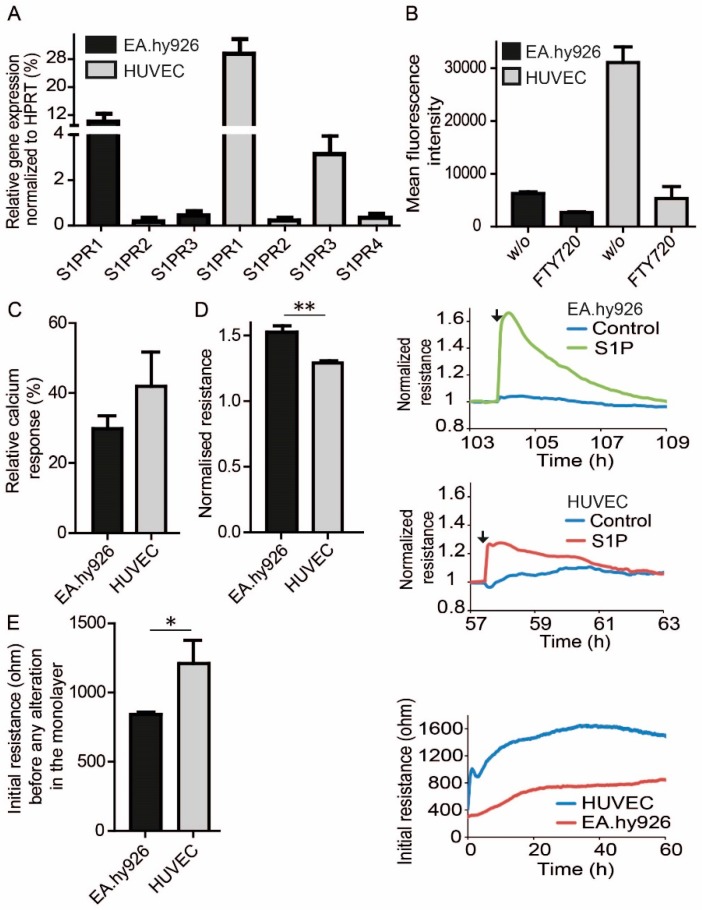
S1PR expression and signaling in EC. (**A**) qPCR analysis of S1PR expression in EA.hy926 and HUVEC. Dara are means ± SEM, *n* = 3. (**B**) Flow Cytometric analysis cell surface expression of S1PR1 on EC before and after treatment with 1 µM FTY720 overnight. means ± SEM, *n* = 3. (**C**) Intracellular calcium responses in EA.hy926 and HUVEC upon stimulation with 100 nM S1P. Data were normalized to the response of 10 µM ATP. Means ± SEM, *n* = 3. (**D**) Resistance following treatment with 1 µM S1P, normalized resistance values were taken at the time of the established maximum resistance after S1P treatment divided by resistance of carrier-treated control cells at the same time and are means ± SEM, *n* = 3, ** *p* <0.01, determined by two-sided Student’s t-test. Line plots represent one experiment out of three with black arrows indicating the addition of S1P or vehicle at the corresponding time. (**E**) Difference in initial non-stimulated resistance of EA.hy926 and HUVEC in ECIS measurements 60 h after seeding, means ± SEM, *n* = 3, * *p* < 0.05, determined by a two-sided Student’s t-test. Line plot represents one experiment out of three.

**Figure 2 cells-09-00928-f002:**
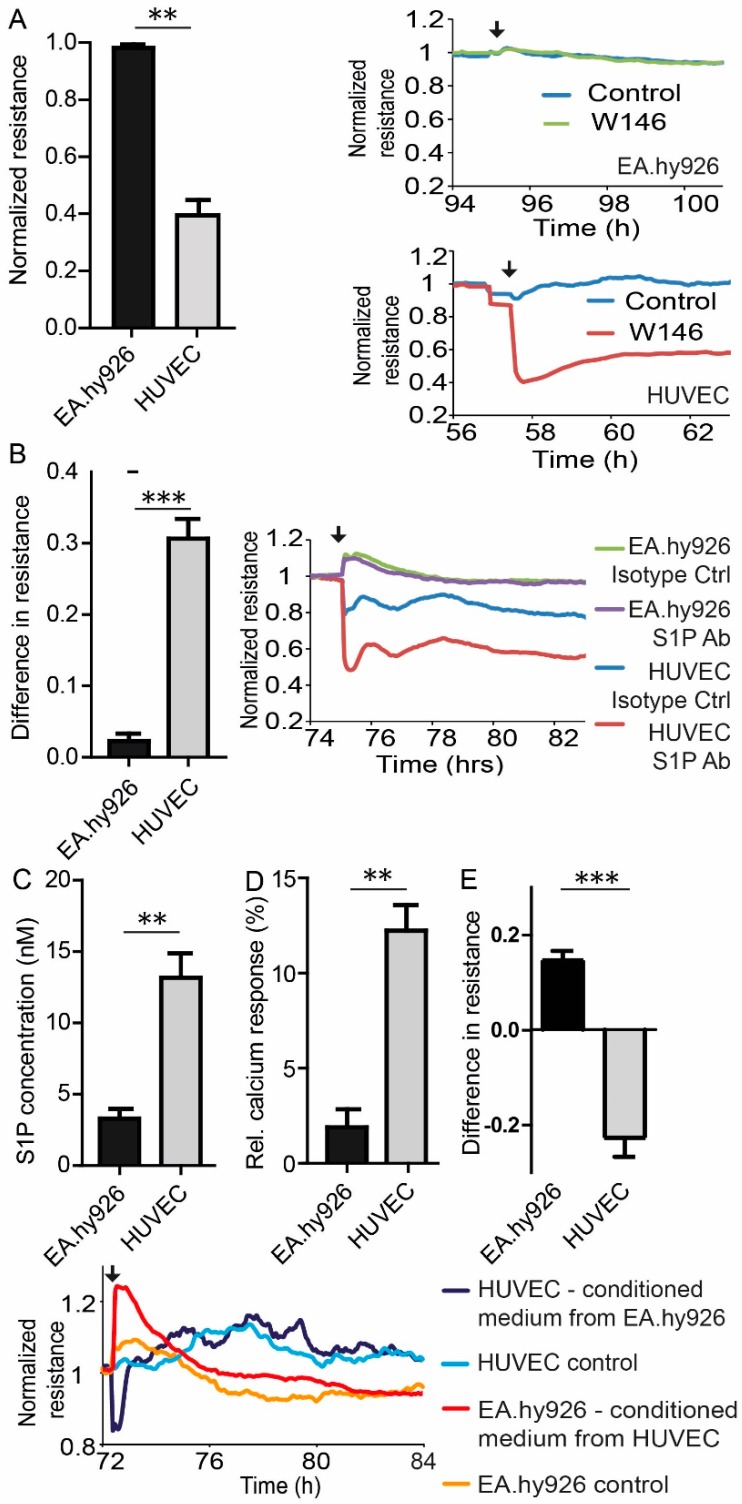
Comparison of S1P-signaling in HUVEC and EA.hy926. (**A**) Resistance following treatment with 3 µM of the S1PR1 antagonist W146. Normalized resistance values were taken at the time of the established maximal change of resistance after W146 treatment divided by resistance of carrier-treated control cells at the same time and are means ± SEM, *n* = 3, ** *p* < 0.001, determined by two-sided Student’s t-test. Line plots represent one experiment out of three with black arrows indicating the addition of W146 or vehicle at the corresponding time. (**B**) Resistance following treatment with 120 µg/mL of the anti-S1P antibody Sphingomab. The difference in resistance is the difference between S1P-antibody treatment and isotype control antibody treatment taken at the time of maximal change of resistance after treatment. Shown are means ± SEM, *n* = 3, *** *p* < 0.001, determined by a two-sided Student’s t-test. Line plot represents one experiment out of three with a black arrow indicating the addition of Sphingomab (S1P Ab) or isotype control antibody (Isotype Ctrl) at the corresponding time. (**C**) LC-MS/MS quantification of extracellular S1P production by EA.hy926 and HUVEC. (**D**) Intracellular calcium response in rat hepatoma HTC4 cells transfected with human S1PR1 and the human Gαi subunit of trimeric G proteins, stimulated with lipid extracts from EA.hy926 and HUVEC as indicated. (**E**) Resistance following medium exchange. The difference in resistance is the difference between exchange of conditioned medium and control medium at the time of maximal change of resistance after medium exchange. Line plot represents one experiment out of three with a black arrow indicating the exchange of medium at the corresponding time, controls represent unconditioned medium. (**C**–**E**) Data are means ± SEM, *n* = 3, ** *p* < 0.01, *** *p* < 0.001, determined by a two-sided Student’s *t*-test.

**Figure 3 cells-09-00928-f003:**
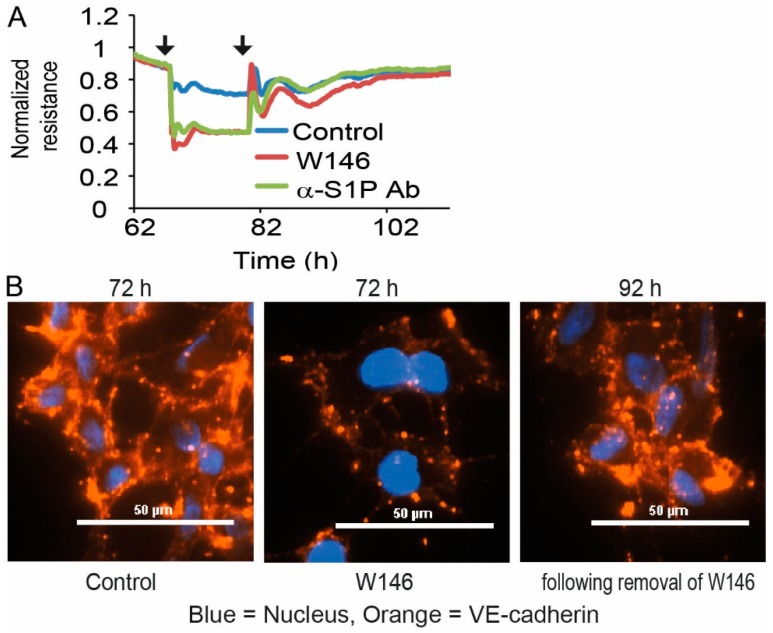
Dependence and reversibility of EC barrier stability in HUVEC and EA.hy926. (**A**) Resistance following treatment with 3 µM S1PR1 antagonist W146 or 120 µg/mL of anti-S1P antibody Sphingomab, followed by the removal of the added substances. Line plot represents one experiment out of three with black arrows indicating the addition and removal of W146 or Sphingomab at the corresponding time points. (**B**) Immunofluorescence staining of VE-cadherin in HUVEC after addition of 3 µM S1PR1 antagonist W146, followed by removal of the added substance. Representative images from one out of three individual experiments are shown. Pictures were taken 6 h after addition of W146 and 12 h following removal of W146.

**Figure 4 cells-09-00928-f004:**
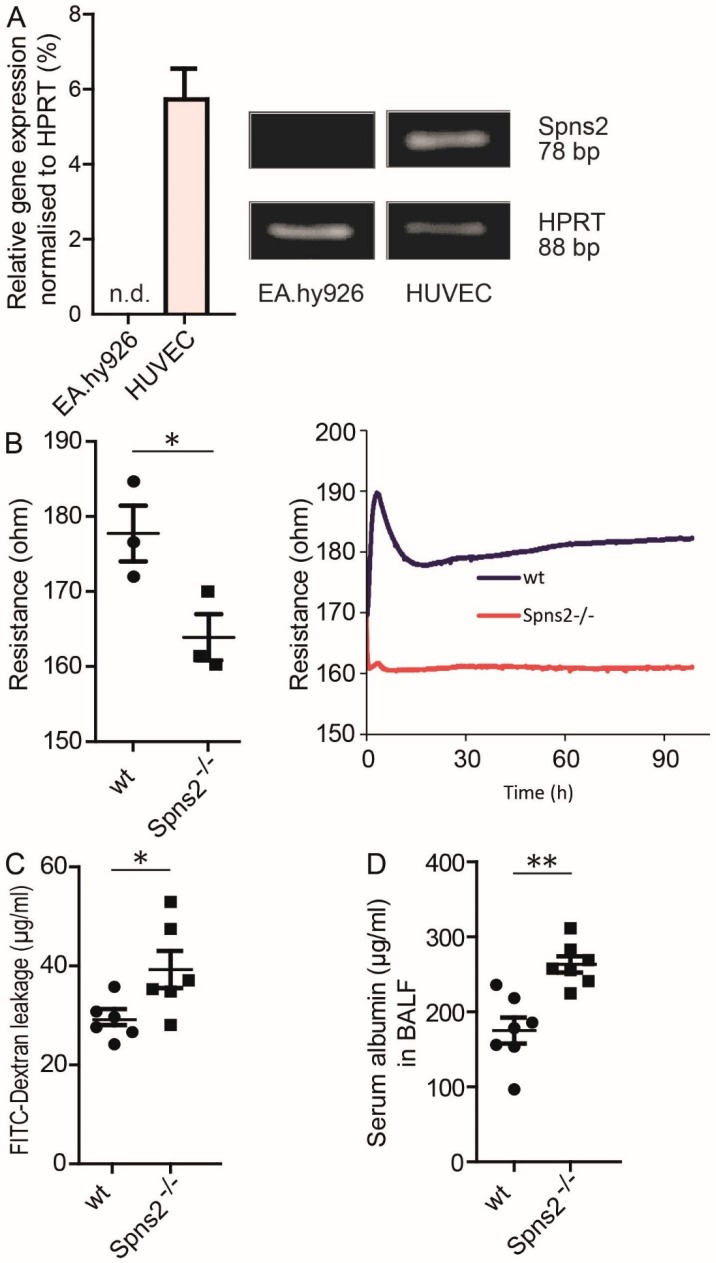
Role of Spns2 for EC barrier stability. (**A**) qPCR analysis of *Spns2* in EA.hy926 and HUVEC, *n* = 3, means ± SEM. Images show representative agarose gel electrophoresis signals of amplified PCR products for *HPRT* and *Spns2*. (**B**) Difference in initial non-stimulated resistance of primary lung ECs isolated from wt and Spns2 deficient mice in ECIS measurements, means ± SEM, *n* = 3, * *p* < 0.05, determined by two-sided Student’s *t*-test. Single values represent the resistance values for separate mice taken 90 h after seeding. Line plot represents one experiment out of three. (**C**) FITC-dextran leakage assay with primary lung endothelial cells isolated from wt and Spns2 deficient mice, means ± SEM, *n* = 6, * *p* < 0.05, determined by two-sided Student’s t-test. (**D**) Serum albumin measurement in BALF isolated from wt and Spns2 deficient mice, means ± SEM, *n* = 6, ** *p* < 0.01, determined by a two-sided Student’s *t*-test.

**Figure 5 cells-09-00928-f005:**
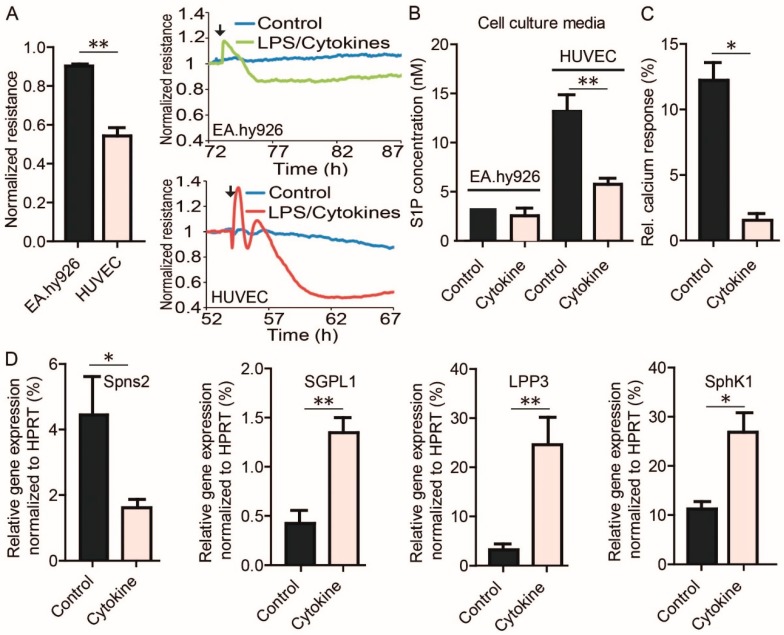
Influence of LPS and cytokines on EC barrier stabilization. (**A**) Resistance following treatment with a mix of LPS and cytokines (50 ng/mL IL1β, 50 ng/mL TNFα, 1 µg/mL LPS). Normalized resistance values are resistance of cells 12 h after LPS/cytokine treatment divided by resistance of carrier-treated control cells at the same time and are means ± SEM, *n* = 3, ** *p* <0.01, determined by a two-sided Student’s *t*-test. Line plots represent one experiment out of three with black arrows indicating the addition of LPS and cytokines or vehicle at the corresponding time. (**B**) LC-MS/MS quantification of extracellular S1P production by EA.hy926 and HUVEC after stimulation with LPS and cytokines or vehicle control. (**C**) Intracellular calcium response in rat hepatoma HTC4 cells transfected with human S1PR1 and the human Gαi subunit of trimeric G proteins, stimulated with lipid extracts from HUVEC without or with LPS and cytokines as indicated. (**D**) qPCR analysis of *Spns2*, *SGPL1*, *LPP3*, and *SphK1* expression in cytokine mix treated HUVEC. (**B**–**D**) Data are means ± SEM, *n* = 3, * *p* < 0.05, ** *p* < 0.01, determined by two-sided Student’s *t*-test.

**Figure 6 cells-09-00928-f006:**
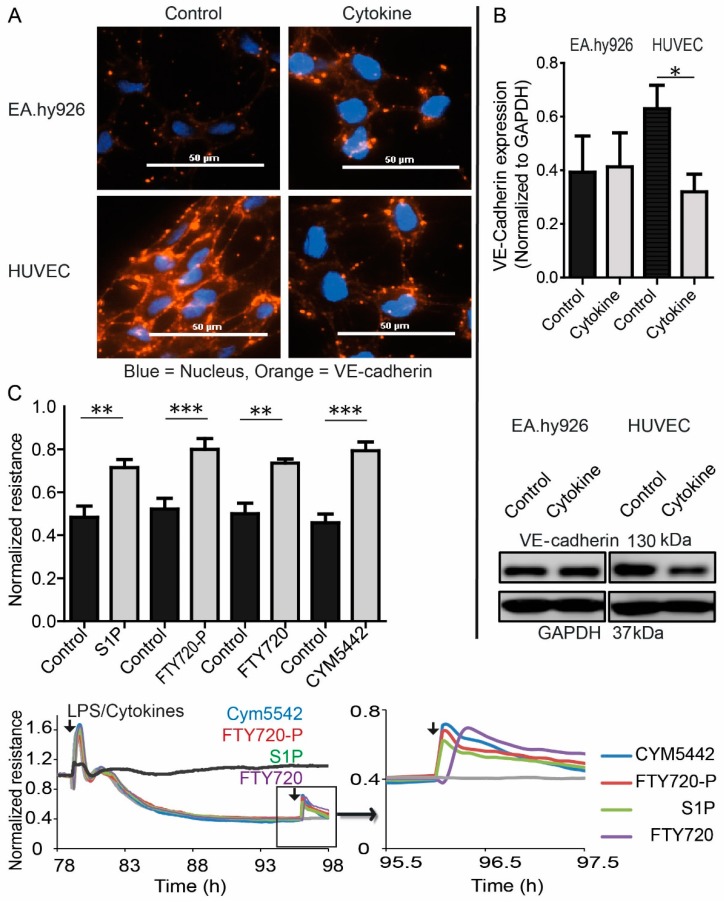
Role of S1P and S1PRs in EC barrier destabilization induced by LPS and cytokines. (**A**) Immunofluorescence staining of VE-cadherin and Western blot analysis of VE-cadherin and GAPDH in EA.hy926 and HUVEC after stimulation with LPS and cytokines or vehicle. Representative images from one out of three individual experiments are shown, size bar = 50 µm. (**B**) Western blot quantification (top) and representative Western blot (bottom) of VE-cadherin expression in EA.hy926 and HUVEC after stimulation with LPS and cytokines or vehicle, means ± SEM, *n* = 3, * *p* < 0.05, determined by two-sided Student’s t-test. (**C**) Resistance following treatment with a mix of LPS and cytokines (50 ng/mL IL1β, 50 ng/mL TNFα, 1 µg/mL LPS), and re-stimulated with 1 µM of the S1PR1 agonist CYM5442, 1 µM of the S1PR1,3,4,5 agonist FTY720-phosphate and its non-phosphorylated precursor FTY720, and 1 µM S1P. Line plots represent one experiment out of three with black arrows indicating the addition of stimuli at the corresponding time points. The dark grey line represents an unstimulated control, the light grey line represents a control stimulated with LPS and cytokines without second stimulation. Bar graph represents means ± SEM, *n* = 3, ** *p* < 0.01, *** *p* < 0.001, determined by two-sided Student’s *t*-test. Normalized resistance values were taken before (controls) and after treatment with S1P, FTY720-P, FTY720, and CYM5442 at the time of the established maximal change of resistance of cells divided by resistance of cells before LPS/cytokine treatment.

**Table 1 cells-09-00928-t001:** Sequences of primers and probes used for quantitative polymerase chain reaction (qPCR) analyses.

Gene	Forward Primer	Reverse Primer	Probe (5′-56-FAM; 3′-36-TAM)
*HPRT*	agcctaagatgagagttc	cacagaactagaacattgata	atctggagtcctattgacatcgcc
*S1PR1*	agcactatatcctcttctg	tgaccaaggagtagattc	tcttcactctgcttctgctctcc
*S1PR2*	catcgtcatcctctgttg	agtggaacttgctgtttc	ccttctggtgctcattgcgg
*S1PR3*	ccaagcagaagtaaatcaag	catggagacgatcagttg	agcagcaacaatagcagccac
*S1PR4*	gcttctgtgtgattctgg	ccatgatcgaacttcaatg	cctctctgggcctcagtagg
*S1PR5*	ggaacaatgatggagatt	ggcattgtccttgataac	attccactcttacactcaattcctgag
*SK1*	ggcagcttccttgaacca	gcaggttcatgggtgaca	ctatgagcaggtcaccaatgaagacctcct
*SK2*	ccgacggcctctcagt	cctggccctgggtctta	acagtgagacctgactccttgctcctacc
*SGPL1*	cctagcacagaccttctgatgt	actccatgcaattagctgcca	aaggcctttgagccctactt
*SPNS2*	ttactggctccagcgtga	tgatcatgcccaggacag	ctgggcattgcgggtgtc

Abbreviations: FAM, 6-carboxyfluorescein; TAM, tetramethylrhodamine.

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
