# Peer review of "Inflammatory Conditions Disrupt Constitutive Endothelial Cell Barrier Stabilization by Alleviating Autonomous Secretion of Sphingosine 1-Phosphate"

_cells, 2020, doi:10.3390/cells9040928_

Round 1
Reviewer 1 Report
The authors present a new mechanism for barrier integrity maintenance. Specifically, they show how the secretion of sphingosine-1-phosphate (SP1) by endothelial cells contributes to maintaining the endothelial barrier via Spns2.
We believe this manuscript is overall a good work of moderate interest, and can be accepted after minor revisions. Specifically, the significance of the study and context is not clear in the abstract, which should be improved. Along the same line, authors need to stress out and clarify in the introduction and discussion the relevance of this mechanism in a biological setting.
Author Response
Dear Editor, dear reviewers,
Thank you very much for your thorough review and your valuable comments, that I like to answer as follows:
Reviewer 1:
The authors present a new mechanism for barrier integrity maintenance. Specifically, they show how the secretion of sphingosine-1-phosphate (SP1) by endothelial cells contributes to maintaining the endothelial barrier via Spns2.
We believe this manuscript is overall a good work of moderate interest, and can be accepted after minor revisions. Specifically, the significance of the study and context is not clear in the abstract, which should be improved. Along the same line, authors need to stress out and clarify in the introduction and discussion the relevance of this mechanism in a biological setting.
Response: Thank you very much for your appreciation. We added the first sentence in the abstract, the last 2 sentences in the introduction and the new chapter “Conclusions” at the end of the discussion to clarify the significance and relevance of this mechanism in a biological setting.
Reviewer 2 Report
Comments:
Abstract:
- Clear concise abstract describing main findings in paper.
- In abstract state that EA.hy926 cells do not export S1P into extracellular space, this contradicts Fig 2C, which indicates low level excretion. Amend accordingly.
- Last sentence too definitive given data, your data suggest this mechanism, but is not definitive i.e. ABC transporters may also be involved. Revise accordingly.
Introduction:
- Line 40: Term ‘permeability’ incorrectly used suggest term ‘movement’
- Sentence starting in line 52 doesn’t make sense, revise.
- A few grammatical errors in introduction, suggest an English check.
- Last para: I think some of the conclusions are a bit too concrete, whilst the authors did show that Spns2 deficient mice have reduced barrier integrity, they didn’t confirm that it was due to lack (as stated) of S1P export. Alter text accordingly.
Sufficiently detailed materials and methods.
Results:
Overall the results section was well written however amendments are required so the data is more accurately interpreted and conveyed. The amendments to be made are as follows:
- Figure legends throughout script require additional detail to describe the data they are showing e.g. in Fig 1B legend it should be stated that assessing cell surface S1PR; in Fig 2 A it should be stated ‘Resistance following treatment with W146’…… and so on for other similar figures.
- In bar graphs need to state exactly at which time point following treatment this data was captured i.e. Fig 1D, 2A, B &E.
- Need to also state time point for Fig 1E (even though no treatment)
- As the controls for each ECIS experiment have slightly different profiles, it would be more accurate to present the normalised treatment resistance data relative to the normalised control data at same time point (this pertains to Figs 1D; 2A, B & E; 5A ).
- Fig 2A: HUVEC SEM on bar chart is small (n=3) when mean is 0.2 and one of the three values is approx. 0.4 as per ECIS graph to right; suggest authors check data.
- Fig 2B: The EA.hy926 data in bar chart does not correlate with data in ECIS plot and the way it was calculated (as described in legend), this needs to be addressed.
- Fig 2E: control ECIS data (i.e. respective unconditioned media) needs to be shown and included in data analysis, before conclusions can be drawn from this data. Especially as different medias used to culture HUVECs and EA.hy926.
- Fig 3: Pattern of VE-cadherin staining consistent with VE-cadherin expression. Change label ‘Media Exchange’ for ‘following removal of W146’. Also need to state experimental time point for each image.
- Fig 3: Authors should have multiple images from the 3 independent expts with which to conduct image analysis, allowing the generation of quantitative data. Alternatively this data could be made more robust by showing additional images for each treatment in supplementary data e.g. 3 additional images per treatment. Or conduct a western similar to 6B to validate.
- Fig 4B: these initial resistance values shown in the graph on the left should be taken from a specific (and stated) time point and should be taken once the barrier has formed and settled. It is not clear when these data points were taken?
- Fig 4C: expand legend on y axis to be more descriptive.
- Fig 5 A: Need to state what time point following treatment was used for data shown in A. Again in A present the normalised treatment resistance data relative to the normalised control data at same time point, as the control profiles differ for each cell line over time.
- Fig 6C: does 1st arrow correspond with cytokine treatment then second arrow with CYM5../FTY…/S1P…., if so please indicate on graphic more clearly.
Good thorough and interesting discussion
Author Response
Dear Editor, dear reviewers,
Thank you very much for your thorough review and your valuable comments, that I like to answer as follows:
Reviewer 2:
Abstract:
- Clear concise abstract describing main findings in paper.
- In abstract state that EA.hy926 cells do not export S1P into extracellular space, this contradicts Fig 2C, which indicates low level excretion. Amend accordingly.
Response: Thank you very much for this information. We changed the abstract saying that S1P secretion was significantly higher in primary human umbilical vein endothelial cells (HUVEC) compared to the endothelial cell line EA.hy926 instead of saying that S1P was secreted by HUVEC, but not by EA.hy926 cells.
- Last sentence too definitive given data, your data suggest this mechanism, but is not definitive i.e. ABC transporters may also be involved. Revise accordingly.
Response: This is correct. We changed the wording to: “Our work suggest a new mechanism […]”. We point out in the last sentence that “secretion of S1P by EC via Spns2 contributed to constitutive EC barrier maintenance […]”. Contribution means that this may not be the only mechanism for S1P secretion.
Introduction:
- Line 40: Term ‘permeability’ incorrectly used suggest term ‘movement’
Response: Done.
- Sentence starting in line 52 doesn’t make sense, revise.
Response: Done.
- A few grammatical errors in introduction, suggest an English check.
Response: Done.
- Last para: I think some of the conclusions are a bit too concrete, whilst the authors did show that Spns2 deficient mice have reduced barrier integrity, they didn’t confirm that it was due to lack (as stated) of S1P export. Alter text accordingly.
Response: We changed the wording to: “[…] decreased exportation of S1P from ECs due to reduced expression of Spns2 may contribute to EC barrier dysfunction during inflammation” instead of: “[…] contributes to EC barrier dysfunction during inflammation” and added “presumably” to the sentence: “Spns2 deficient mice demonstrated significantly reduced EC barrier formation presumably due to the lack of S1P exportation from ECs.”
Sufficiently detailed materials and methods.
Results:
Overall the results section was well written however amendments are required so the data is more accurately interpreted and conveyed. The amendments to be made are as follows:
- Figure legends throughout script require additional detail to describe the data they are showing e.g. in Fig 1B legend it should be stated that assessing cell surface S1PR; in Fig 2 A it should be stated ‘Resistance following treatment with W146’…… and so on for other similar figures.
Response: Done.
- In bar graphs need to state exactly at which time point following treatment this data was captured i.e. Fig 1D, 2A, B &E.
Response: The exact time point for each experiment was added, e.g. “at the time of the established maximum resistance”.
- Need to also state time point for Fig 1E (even though no treatment)
Response: Done (60 h after seeding).
- As the controls for each ECIS experiment have slightly different profiles, it would be more accurate to present the normalised treatment resistance data relative to the normalised control data at same time point (this pertains to Figs 1D; 2A, B & E; 5A).
Response: We are grateful for this comment because this is, in fact, the way the data were normalized in most cases. We made a mistake stating that “normalized resistance values are maximum resistance after S1P treatment divided by resistance prior to treatment”, which was only the case for Fig. 6C. In all other experiments we used the normalized control data at the same time point. We changed all figure legends accordingly and apologize for this mistake.
- Fig 2A: HUVEC SEM on bar chart is small (n=3) when mean is 0.2 and one of the three values is approx. 0.4 as per ECIS graph to right; suggest authors check data.
Response: We are grateful for this comment. We made a mistake during normalization of the data, which resulted in a more pronounced difference than it actually was. We corrected the bar graph and the statistics accordingly.
- Fig 2B: The EA.hy926 data in bar chart does not correlate with data in ECIS plot and the way it was calculated (as described in legend), this needs to be addressed.
As mentioned earlier, data in Fig. 2B were normalized with the control data and not, as stated before, with the data before addition of the antibody. The difference in resistance is therefore the difference between S1P-antibody treatment and isotype control antibody treatment. The figure legend was changed accordingly.
- Fig 2E: control ECIS data (i.e. respective unconditioned media) needs to be shown and included in data analysis, before conclusions can be drawn from this data. Especially as different medias used to culture HUVECs and EA.hy926.
Response: The control ECIS data are now included.
- Fig 3: Pattern of VE-cadherin staining consistent with VE-cadherin expression. Change label ‘Media Exchange’ for ‘following removal of W146’. Also need to state experimental time point for each image.
Response: The labeling was changed accordingly, and the time points were added in the figure legend.
- Fig 3: Authors should have multiple images from the 3 independent expts with which to conduct image analysis, allowing the generation of quantitative data. Alternatively this data could be made more robust by showing additional images for each treatment in supplementary data e.g. 3 additional images per treatment. Or conduct a western similar to 6B to validate.
Response: We added additional pictures in supplemental figure S1.
- Fig 4B: these initial resistance values shown in the graph on the left should be taken from a specific (and stated) time point and should be taken once the barrier has formed and settled. It is not clear when these data points were taken?
Response: Data points represent the mean of resistance values for separate mice taken every 10 min between 10 and 100 min. We added this information to the figure legend. We also realized that we took the wrong data (wrong frequency) for the plot. We corrected the plot accordingly.
- Fig 4C: expand legend on y axis to be more descriptive.
Response: Done (FITC-dextran leakage).
- Fig 5 A: Need to state what time point following treatment was used for data shown in A. Again in A present the normalised treatment resistance data relative to the normalised control data at same time point, as the control profiles differ for each cell line over time.
Response: The timepoint is now mentioned in the figure legend (12 h after LPS/cytokine treatment). As mentioned before, the normalized treatment resistance data relative to the normalized control data at the same time are shown, which is now mentioned in the figure legend.
- Fig 6C: does 1starrow correspond with cytokine treatment then second arrow with CYM5../FTY…/S1P…., if so please indicate on graphic more clearly.
Response: Done.
Good thorough and interesting discussion
Reviewer 3 Report
Manuscript by Jefri et al. talks about important role of spns2 in maintaining the steady release of
S1P from endothelial cells. Role of spns2 is very well known in leukocyte infiltration and targeting
spns2 is an effective way to control the immune responses in various diseases.
Major concern in the manuscript;
1. Authors should bring a clear point in their conclusion about spns2 in septic shock. Pathology of
sepsis includes increased systemic inflammatory response and endothelial permeability. In such
conditions, targeting spns2 will be useful to limit inflammatory response. Is author is trying to say
that targeting spns2 also cause a side effect in form of edema or increased endothelial permeability
in sepsis?
2. Line 181, BALF was diluted 1:20.000; Is it 1:20 or 1:20,000?
3. V.E cadherin staining is not good in the figures. It should be done on confluent cells. It usually
comes as a tiles on the floor like structure in immunostaining. However, I cannot see that.
4. What time point was chosen for plotting quantification for ECIS experiments?
5. Figure 2A: Why W146 was added after 95 hours in EA.hy926 cells, while in HUVECs it was
added at 57 hours. At later time point, EA.hy926 may be too much confluent, that why author
might not see any effect. (EA.hy926 can grow in double layers).
6. Figure 2B: why Y axis plotted as difference in resistance. It can be plotted as simply as other
that is normalized resistance with four bars to avoid confusion.
7. Figure 3B: Magnification of W146 treated figures looks different (nucleus is bigger). Although
scale bars shown are similar.
8. Figure 4B line graph; Do spns-/- cells form a good junction. Immunostaining for VE cadherin
should be performed on these cells to conclusively say that whether attachment is a problem or
junction integrity.
9. Figure 4B line graph; It’s difficult to understand the line curves. After seeding cell curve should
start from 0 hrs and then it should increase. WT type cells also do not show this curve. Curve
should be like in figure 1E where EAhy926 cell and HUVECs shows steady increase in resistance
over the time. Hence, it looks like spns2 KO cells have attachment problem as well as cell growth
problem. In either case there will be an increase in permeability. Author should discuss about this.
10. Figure 6A: Why EAhy926 does not show any VE-cadherin staining. Also, the magnification
of control figures and treated figures looks different (nucleus are bigger). What was the passage of
EA.hy926 cells used in these experiments? Increased passage leads to loss of endothelial
phenotype in EA.hy926 cells.
11. Figure 6B: It should include a representative western blot since immunostaining of VE
cadherin does not show up anything in EA.hy926 cells.
12. Figure 6C is missing control line curve from ECIS picture.
Minor corrections;
References are numbered twice. Please correct it.
Author Response
Dear Editor, dear reviewers,
Thank you very much for your thorough review and your valuable comments, that I like to answer as follows:
Reviewer 3:
Manuscript by Jefri et al. talks about important role of spns2 in maintaining the steady release of
S1P from endothelial cells. Role of spns2 is very well known in leukocyte infiltration and targeting
spns2 is an effective way to control the immune responses in various diseases.
Major concern in the manuscript;
1. Authors should bring a clear point in their conclusion about spns2 in septic shock. Pathology of
sepsis includes increased systemic inflammatory response and endothelial permeability. In such
conditions, targeting spns2 will be useful to limit inflammatory response. Is author is trying to say
that targeting spns2 also cause a side effect in form of edema or increased endothelial permeability
in sepsis?
Response: Spns2 is a difficult target as a transporter, particularly when it is downregulated during infections. But the observed local paracrine and autocrine activity of constitutively transported S1P may provide the chance to locally supplement with S1P or S1PR1 agonists, e.g. via constant inhalation in lung, or to overcome the reduced S1P exportation with increased S1P accumulation in tissue cells, e.g. by inhibition of the S1P-degrading enzyme S1P-lyase. We added this conclusion as an additional chapter on page 14.
Line 181, BALF was diluted 1:20.000; Is it 1:20 or 1:20,000?
Response: 1:20,000 is correct, the ELISA was very sensitive.
V.E cadherin staining is not good in the figures. It should be done on confluent cells. It usually
comes as a tiles on the floor like structure in immunostaining. However, I cannot see that.
Response: We added additional pictures in supplemental figure S1 to make it clearer.
What time point was chosen for plotting quantification for ECIS experiments?
Response: Time points are now added to the figure legends. In general, data were taken at the time of the maximal response after treatment, if not otherwise stated.
Figure 2A: Why W146 was added after 95 hours in EA.hy926 cells, while in HUVECs it was
added at 57 hours. At later time point, EA.hy926 may be too much confluent, that why author
might not see any effect. (EA.hy926 can grow in double layers).
Response: EA.hy926 cells required a longer period of time to establish a stable base line with constant resistance values that were not rising anymore. At that stage, cells responded very well to S1P as a barrier-enhancing stimulus. Therefore, we are sure that the cells were still in good shape and not overgrown.
Figure 2B: why Y axis plotted as difference in resistance. It can be plotted as simply as other
that is normalized resistance with four bars to avoid confusion.
Response: We thought about plotting the resistance values for controls and treated cells separately, but decided to plot the difference because of the fact that particularly HUVEC also responded to the control isotype antibody treatment to some extent unspecifically, which can be seen in the representative original line graph. To avoid confusion with uneven EA.hy926 and HUVEC controls, we decided to normalize the response to the control treatment values by plotting the difference.
Figure 3B: Magnification of W146 treated figures looks different (nucleus is bigger). Although
scale bars shown are similar.
Response: Indeed, the nuclei appear bigger at the same magnification. One reason could be increased appearance of euchromatin due to cell activation or cell stress. This, however, is just speculation. The magnification remained the same.
Figure 4B line graph; Do spns-/- cells form a good junction. Immunostaining for VE cadherin
should be performed on these cells to conclusively say that whether attachment is a problem or
junction integrity.
Response: This would certainly be a good idea. Unfortunately, the yield of primary lung endothelial cells from single mice was sufficient to do resistance measurements and to test FITC-dextran leakage, but didn’t allow for additional assays or Western blot analyses. However, both, resistance measurements and FITC-dextran leakage provided similar results demonstrating a decreased barrier stability of lung EC from Spns2 knockout mice. In addition, direct measurement of albumin in BALF from wt and Spns2 knockout mice supported this conclusion. Of course, we also checked for cell attachment by microscopy after cell isolation, which was similar for isolated lung EC from wt and Spns2 knockout mice.
Figure 4B line graph; It’s difficult to understand the line curves. After seeding cell curve should
start from 0 hrs and then it should increase. WT type cells also do not show this curve. Curve
should be like in figure 1E where EAhy926 cell and HUVECs shows steady increase in resistance
over the time. Hence, it looks like spns2 KO cells have attachment problem as well as cell growth
problem. In either case there will be an increase in permeability. Author should discuss about this.
Response: Isolated primary lung EC had a growth arrest, so that we seeded them with quite high cell numbers to compensate for this effect and to be able to establish a confluent cell layer. A second reason for the different resistance curve could be the coating of the cell chambers, which may have initially induced a certain background resistance. Coating, however, was necessary for a reliable cell attachment. We added these aspects to the discussion (page 13).
Figure 6A: Why EAhy926 does not show any VE-cadherin staining. Also, the magnification
of control figures and treated figures looks different (nucleus are bigger). What was the passage of
EA.hy926 cells used in these experiments? Increased passage leads to loss of endothelial
phenotype in EA.hy926 cells.
Response: As mentioned in the response to comment 7, the nuclei appear bigger at the same magnification. One reason could be increased appearance of euchromatin due to cell activation or cell stress. This, however, is just speculation. The magnification remained the same. EA.hy926 cells were used between passage 4-15 in experiments from our original stock. We did not observe any phenotypical changes between these passages. VE-cadherin staining was indeed much weaker compared to HUVEC, which was also reflected in Western blot analyses (Fig. 6B), although VE-cadherin was certainly well expressed.
Figure 6B: It should include a representative western blot since immunostaining of VE
cadherin does not show up anything in EA.hy926 cells.
Response: A representative Western blot was added.
Figure 6C is missing control line curve from ECIS picture.
Response: The control values shown in the bar graph correspond to resistance values right before stimulation, they do not derive from additional controls. We are sorry about this confusion and added this explanation to the figure legend.
Minor corrections;
References are numbered twice. Please correct it.
Response: We double-checked the reference list, but could not find a mistake. Is it possible that numbering of the references was mixed up with the numbering of lines? – If the problem still exists, please specify, we are happy to correct it.
Round 2
Reviewer 2 Report
What the authors have done in Fig 2B is not apparent in the text, for clarification the Fig 2B legend needs to state ‘The difference in resistance is the difference between S1P-antibody treatment and isotype control antibody treatment.’
Fig 2E, despite control data being shown it has not been taken into account for data analysis as requested in my original rev. Furthermore as the controls for each ECIS experiment have slightly different profiles, it would be more accurate to present the normalised treatment resistance data relative to the normalised control data at same time point (again as requested in my original review for this figure).
Fig 4B. My original comment:
‘Fig 4B: these initial resistance values shown in the graph on the left should be taken from a specific (and stated) time point and should be taken once the barrier has formed and settled. It is not clear when these data points were taken?’
This comment was not addressed; the authors response to this comment indicates that the graph on the left shows data from many time points between 10min-100min, which includes the period during which barrier is being established.
Author Response
Dear Editor, dear reviewers,
Thank you very much again for your thorough review and your valuable comments, that I like to answer as follows:
Reviewer 2:
What the authors have done in Fig 2B is not apparent in the text, for clarification the Fig 2B legend needs to state ‘The difference in resistance is the difference between S1P-antibody treatment and isotype control antibody treatment.’
Response: Thank you very much for this valuable comment. We changed the figure legend accordingly.
Fig 2E, despite control data being shown it has not been taken into account for data analysis as requested in my original rev. Furthermore as the controls for each ECIS experiment have slightly different profiles, it would be more accurate to present the normalised treatment resistance data relative to the normalised control data at same time point (again as requested in my original review for this figure).
Response: Thank you very much for this clarification. We now changed the bar graph showing the difference in resistance between exchange of control media and conditioned media. We also changed the figure legend accordingly.
Fig 4B. My original comment:
‘Fig 4B: these initial resistance values shown in the graph on the left should be taken from a specific (and stated) time point and should be taken once the barrier has formed and settled. It is not clear when these data points were taken?’
This comment was not addressed; the authors response to this comment indicates that the graph on the left shows data from many time points between 10min-100min, which includes the period during which barrier is being established.
Response: We are sorry for not taking care of it initially. We now changed the figure and took the resistance values of each single mouse 90 h after seeding. At this timepoint the barrier was well established as seen in the line graph.
Reviewer 3 Report
Authors have improved the manuscript from the earlier version. However, I still see some minor errors that needs to be fixed;
Page 8 of 17, Figure legend; B: spell mistake: Resistance.
Page 13 of 17, Please remove this line: A second reason could be an initial background resistance that may derive from the coating of the cell chambers, which was required for cell attachment.
This cannot be a reason since initial background resistance must be same for both WT and KO. However, I agree with other reason that cells might be growing less as reported by others in retina (Chao Fang, S1P transporter SPNS2 regulates proper postnatal retinal morphogenesis, FASEB J. 2018 Jul;32(7):3597-3613.).
Question to a response (Figure 6C): The control values shown in the bar graph correspond to resistance values right before stimulation, they do not derive from additional controls. We are sorry about this confusion and added this explanation to the figure legend.
I guess author's didn't understand my question. I mean to say that control line is missing in the line graph at two treatments.
- 1st treatment: CTL line (untreated i.e., without LPS/cytokine) should be added same as in line graph of Figure 5A (HUVECs) Blue line.
- 2nd treatment : LPS/cytokine line (untreated for 2nd treatment) is missing.
Author Response
Dear Editor, dear reviewers,
Thank you very much again for your thorough review and your valuable comments, that I like to answer as follows:
Reviewer 3:
Authors have improved the manuscript from the earlier version. However, I still see some minor errors that needs to be fixed;
Page 8 of 17, Figure legend; B: spell mistake: Resistance.
Response: Corrected.
Page 13 of 17, Please remove this line: A second reason could be an initial background resistance that may derive from the coating of the cell chambers, which was required for cell attachment.
This cannot be a reason since initial background resistance must be same for both WT and KO. However, I agree with other reason that cells might be growing less as reported by others in retina (Chao Fang, S1P transporter SPNS2 regulates proper postnatal retinal morphogenesis, FASEB J. 2018 Jul;32(7):3597-3613.).
Response: Thank you very much for sharing this information. We deleted the mentioned sentence accordingly.
Question to a response (Figure 6C): The control values shown in the bar graph correspond to resistance values right before stimulation, they do not derive from additional controls. We are sorry about this confusion and added this explanation to the figure legend.
I guess author's didn't understand my question. I mean to say that control line is missing in the line graph at two treatments.
- 1st treatment: CTL line (untreated i.e., without LPS/cytokine) should be added same as in line graph of Figure 5A (HUVECs) Blue line.
- 2nd treatment : LPS/cytokine line (untreated for 2nd treatment) is missing.
Response: We are sorry for not getting this right initially. We now added the suggested controls to the figure. Due to space limitations, the two control lines are explained in the figure legend.